# Folding Dynamics of 3,4,3-LI(1,2-HOPO) in Its Free and Bound State with U^4+^ Implicated by MD Simulations

**DOI:** 10.3390/molecules27238151

**Published:** 2022-11-23

**Authors:** Qin Wang, Ziyi Liu, Yu-Fei Song, Dongqi Wang

**Affiliations:** 1State Key Laboratory of Chemical Resource Engineering, Beijing Advanced Innovation Center for Soft Matter Science and Engineering, Beijing University of Chemical Technology, Beijing 100029, China; 2State Key Laboratory of Fine Chemicals, Liaoning Key Laboratory for Catalytic Conversion of Carbon Resources, School of Chemical Engineering, Dalian University of Technology, Dalian 116024, China; 3CAS-HKU Joint Laboratory of Metallomics on Health and Environment, Multidisciplinary Initiative Center, Institute of High Energy Physics, Chinese Academy of Sciences, Beijing 100049, China

**Keywords:** 3,4,3-LI(1,2-HOPO), molecular dynamics, decorporation, actinides, tetravalent uranium ion

## Abstract

The octadentate hydroxypyridonate ligand 3,4,3-LI(1,2-HOPO) (*t*-HOPO) shows strong binding affinity with actinide cations and is considered as a promising decorporation agent used to eliminate in vivo actinides, while its dynamics in its unbound and bound states in the condensed phase remain unclear. In this work, by means of MD simulations, the folding dynamics of intact *t*-HOP^O^ in its neutral (*t*-HOPO0) and in its deprotonated state (*t*-HOPO^4^*^−^*) were studied. The results indicated that the deprotonation of *t*-HOPO in the aqueous phase significantly narrowed the accessible conformational space under the simulated conditions, and it was prepared in a conformation that could conveniently clamp the cations. The simulation of U^IV^-*t*-HOPO showed that the tetravalent uranium ion was deca-coordinated with eight ligating O atoms from the *t*-HOPO^4^*^−^* ligand, and two from aqua ligands. The strong electrostatic interaction between the U^4+^ ion and *t*-HOPO^4^*^−^* further diminished the flexibility of *t*-HOPO^4^*^−^* and confined it in a limited conformational space. The strong interaction between the U4+ ion and *t*-HOPO^4^*^−^* was also implicated in the shortened residence time of water molecules.

## 1. Introduction

The accumulation of radioactive metals in organisms causes damage to tissues and organs owing to their chemical and radio toxicity [1,2]. In order to decorporate in vivo radioactive metal ions, a few chelating agents have been proposed in the past few decades [3,4]. Owing to complex biological conditions, e.g., variations in pathology and the redox of metal ions, the development of decorporation agents turned out to be challenging [5], and the earlier reported ethylenediaminetetraacetic acid (EDTA) [6] was deprecated due to its high affinity with divalent metal ions [7]. EDTA was first used as an analytical agent for calcium due to its high affinity for Ca(II) (logK_CaL_ = 10.28) in 1947 [8]. Later studies have shown that it is able to remove in vivo ^239^Pu and ^241^Am, as it benefitted from its high complexation ability toward Pu [9]. Diethylenetriaminepentaacetic acid (DTPA) was reported in 1954 as an alternative to EDTA. It shows good selectivity between divalent metal ions and tri-/tetravalent actinides [10], and its salts have been approved for clinic use as actinide chelators. However, it is flawed in its, e.g., insufficient sequestering strength, and it has low efficiency for the removal of hydrolyzed or intracellular actinides [11,12]. Ideal decorporation agents are expected to meet the requirements of free of toxicity, high selectivity, and strong binding strength.

Stimulated by siderophores, which are secreted by microorganisms and responsible for iron trafficking from the host owing to their high iron affinity [13,14], many sequestering agents have been developed to target actinides. These compounds were divided into five categories by Raymond et al. [14]. according to their functional group: (1) catecholamide (CAM) ligands; (2) terephthalamide (TAM) ligands; (3) hydroxypyridinone (HOPO) ligands; (4) mixed ligands; and (5) sulfonamide ligands. The development of the actinide decorporation agents has been reviewed recently [15]. Among these ligands, HOPOs, which coin a hydroxyl and a keto group on a hexa-membered azaheterocycle and are classified into three classes, i.e., 1-hydroxypyridin-2-one (1,2-HOPO), 3-hydroxypyridin-2-one (3,2-HOPO), and 3-hydroxypyridin-4-one (3,4-HOPO) [16], have been used as building blocks to develop a large number of chelating agents. Their in vivo decorporation efficiency for actinides has been evaluated, and the octadentate hydroxypyridinonate ligand, 3,4,3-LI(1,2-HOPO), denoted here as *t*-HOPO, is one of the most promising candidates for new decorporation therapies. The *t*-HOPO exhibits high affinity and selectivity [17] to bind with thorium [18], uranium [19], neptunium [20], plutonium [21,22,23], americium [20], curium [20,24], berkelium [25], californium [25] and lanthanides [26], and is regarded as a better actinide decorporation agent than DTPA [20]. Recently, Abergel and coworkers [21] measured the stability constants of tetravalent plutonium and thorium complexed with *t*-HOPO, and Yang and coworkers [27] reported a DFT study to reveal the bond property between *t*-HOPO and tri-/tetravalent actinides.

These studies improved our understanding of the interaction modes between *t*-HOPO and actinides, while the dynamics of *t*-HOPO and its complexation with actinides in the aqueous phase remain unclear. This motivated the present work in which we applied molecular dynamics simulations to study the folding equilibria of *t*-HOPO in its neutral and deprotonated states in aqueous solution. Its complexation with tetravalent uranium (U^4+^) was also investigated here to offer insights on the influence of its complexation with monatomic cations in the aqueous phase. As the most important naturally occurring actinide and a global environmental contaminant, uranium was identified to exist mainly in tetra- and hexa-valent states in the oxidative environment. In recent studies, U^4+^ has been shown to be stable under physiologic conditions, and its coordination with *t*-HOPO enhances its stability in the aqueous phase [28], while we lack understanding of the coordination mode and dynamics of the complex at the molecular level. In this work, choosing the U^IV^(*t*-HOPO) complex as an exemplar enabled us not only to address the influence of the coordination on the dynamics of *t*-HOPO, but also shed light on the migration of in vivo U^4+^ captured by *t*-HOPO.

## 2. Results and Discussion

### 2.1. HOPO in Its Neutral and Deprotonated States

Neutral *t*-HOPO. Both model systems with *t*-HOPO in its neutral (*t*-HOPO^0^) and deprotonated (*t*-HOPO^4−^) states solvated in the aqueous phase (compositions and system sizes are shown in Appendix A) were sampled for 100 ns. During the whole plain MD simulation, the conformation of the neutral *t*-HOPO molecule kept fluctuating, showing that under the simulation conditions, the uncharged *t*-HOPO could facilely visit broader conformational space owing to the competition between the intramolecular interaction and the interaction between *t*-HOPO and solvent. In view of the atomic positional RMSD, as shown in Figure 1a, *t*-HOPO displayed its flexibility with the RMSD in the range of 0.3–0.6 nm. The central structures of the top three clusters obtained from the clustering analysis of simulation trajectories are collected in Figure 2. These clusters covered more than 87% of the trajectories, and they differed from each other in the relative positions of the four *m*-HOPO rings. According to clustering analysis (cutoff = 0.35 nm), the major cluster (cluster 1 in Figure 2) had a population percentage of 59%.

In the neutral state of *t*-HOPO, the most populated cluster (cluster 1, Figure 2) is featured by the mutually paralleled Plane1 with Plane2, and Plane3 with Plane4. In the second and third most populated clusters (clusters 2 and 3, Figure 2) of neutral *t*-HOPO, such a parallel relationship was lost.

The time evolution of the angles between Plane1 and Plane2 (*α*_12_, see Figure 1 for definitions of the four planes), between Plane3 and Plane4 (*α*_34_) and between Plane2 and Plane3 (*α*_23_) were analyzed and are shown in Figure 3a and Appendix A. The *α*_12_ and *α*_34_ angles fluctuated from 0° (parallel) to 180° (antiparallel), indicating that the two pairs of planes could adopt both conformations parallel or antiparallel to each other with more preference for their parallel conformation.

Deprotonated *t*-HOPO^4−^. In the deprotonated *t*-HOPO (denoted as *t*-HOPO^4−^), each *m*-HOPO ring carried −1e charge. These negative charges were neutralized by Na^+^ counterions in the system. The far fewer fluctuations in the atom positional RMSD (Figure 1b) compared to neutral *t*-HOPO (Figure 1a) indicated the stronger conformational rigidity of *t*-HOPO^4−^. The clustering analysis of *t*-HOPO^4−^ used a smaller cutoff (0.18 nm) to obtain a similar population of the largest cluster (cluster 1) as its neutral counterpart (Figure 2). In the top three most populated clusters, Plane1 was parallel to Plane2, and Plane3 was parallel to Plane4. Meanwhile, in view of the time evolution of the angles between the four *m*-HOPO rings in Figure 3b, the four planes showed high propensity to stay parallel in pairs during the simulations. These results indicated that the deprotonated *t*-HOPO^4−^ displayed lower flexibility than the neutral *t*-HOPO during the 100 ns simulations.

IGM analysis. To understand the intramolecular non-covalent interactions in neutral *t*-HOPO and deprotonated *t*-HOPO^4−^, independent gradient model (IGM) analysis was carried out for the major cluster (Cluster 1). As shown in Figure 4, there was an interaction region between Plane1 and Plane2 and between Plane3 and Plane4 in both neutral *t*-HOPO and deprotonated *t*-HOPO^4−^, which was characterized as π-π parallel-displaced stacking interactions according to the IGM analysis. Such π-π parallel-displaced stacking interactions were not identified in the other clusters owing to the longer distances between the planes.

The analysis of hydrogen bonds (H-bonds, Figure 5) showed that there were fewer H-bonds between the four *m*-HOPO rings of *t*-HOPO than those of *t*-HOPO^4−^, and there were fewer H-bonds between *t*-HOPO^0^ and the water solvent than those between *t*-HOPO^4−^ and the solvent. This is consistent with the fewer fluctuations in the RMSD of the latter than the former, indicating that *t*-HOPO^4−^ benefited more from the intra- and inter- H-bonds than *t*-HOPO^0^ in maintaining their favorable conformations during the simulations (Figure 2).

Another beneficial factor for *t*-HOPO^4−^ to maintain its favorable conformation was contributed by the Na^+^ counterion, which was not present in the simulations of *t*-HOPO^0^. During the 100 ns simulation, there was always one Na^+^ ion in the vicinity of each pair of mutually paralleled *m*-HOPO rings of *t*-HOPO^4−^ (Figure 3c). These Na^+^ ions locked the two *m*-HOPO rings via strong electrostatic attractive interactions to maintain the parallel conformation of *t*-HOPO^4−^ (Figure 6 and Appendix A). The analysis of the trajectory indicated that the Na^+^ ions may have occasionally moved away from the *m*-HOPO units, which was accompanied by the disruption of the parallel relationship of the two *m*-HOPO they locked. These wandering Na^+^ ions did not move far away, but migrated back quickly (Figure 7), driven by the electrostatic attraction between them and the *m*-HOPO rings, to recover the parallel relationship between the pair of *m*-HOPO rings.

### 2.2. U^Ⅳ^-t-HOPO Complexes

The *t*-HOPO was reported with high affinity for tetravalent uranium. Here, the binding of HOPO with U^4+^ was investigated to analyze the solution behavior of the complex.

As shown in Figure 8, U^4+^ coordinated with eight oxygen atoms from the four *m*-HOPO units of *t*-HOPO^4−^, with each *m*-HOPO unit offering one hydroxyl oxygen (O^h^) and one carbonyl oxygen (O^c^) on the *m*-HOPO ring (Figure 1), which agreed well with previous studies [27]. During simulations, an exchange of the ligating O^c^ atom on the second *m*-HOPO ring by the carbonyl O of the amide linker (O^a^) was observed (Appendix A). It could be also noticed that there were two water molecules in the first coordination shell of U^4+^ during the simulation. This complemented the very limited number of previous studies in which this was ambiguous in the presence of water ligands in the first coordination shell of actinides when coordinating with *t*-HOPO^4−^ ligand. In an earlier study by Abergel and coworkers [24], the measurement of the luminescence lifetime of the Cm^3+^-*t*-HOPO complex indicated the possible presence of inner-sphere water. In a later computational study, Yang and coworkers [27] reported that the coordination of inner-sphere water to Am(III) in an [Am(*t*-HOPO)]^-^ complex could not survive geometric optimization unless a distance restraint was imposed.

The elusive coordination interaction between actinide ion and ligating water molecules in the [U(*t*-HOPO)]^0^ complexes reported in previous experimental and computational studies [24,27] could be due to the labile nature of the aqua ligands and the underestimation of the solvent effect when handled by the implicit continuum model. This was demonstrated in the present study in which we observed the exchange of the two water molecules in the first shell of U^4+^ with the bulk, showing that the inner-sphere water molecules were in a dynamic equilibrium (Figure 9). We note that in an earlier study of the complexation of uranyl with *t*-HOPO^4−^, Den Auwer and Vidaud and coworkers [29] reported an energy difference of 0.32 eV between the two coordination patterns [UO_2_(κ^8^-*t*-HOPO)]^2−^∙(H_2_O)_2_ and [UO_2_(κ^6^-*t*-HOPO)(H_2_O)_2_]^2−^, which showed the competing coordination of *t*-HOPO and aqua ligands with actinides.

In the [U^IV^-*t*-HOPO]^0^ complex, each [*m*-HOPO]^-^ unit offered its two oxygen atoms (O^h^ and O^c^ in Figure 1) on the ring to coordinate with U^4+^. The coordination mode and key bond distances are shown in Table 1 and Figure 8. The carbonyl oxygen (O^c^), which benefited from its more negative atomic charge than the deprotonated O^h^ site, built stronger interaction with U, as seen in the shorter U^IV^-O^c^ distance than that of U^IV^-O^h^.

Upon binding with U^4+^, the *t*-HOPO^4−^ became more compact in view of its smaller radius of gyration (Rg) (Figure 10a) and its smaller eccentricity (Figure 10b). This agrees with a much more limited conformational space in the *θ*_1_-*θ*_2_ map of the U^IV^-*t*-HOPO complex than that of *t*-HOPO^4−^ (Figure 11). In *t*-HOPO^4−^, *θ*_1_ and *θ*_2_ visited broader space than that of the U^IV^-*t*-HOPO complex in which the population of two angles was confined in the region of around (100°,100°). The fluctuations of *θ*_1_ and *θ*_2_ were directly relevant to the conformational stability of the *t*-HOPO^4−^ ligand; thus, the smaller fluctuations of *θ*_1_ and *θ*_2_ in the U^IV^-*t*-HOPO than those in the free *t*-HOPO^4−^ indicated the significantly narrowed conformational space of the ligand upon its coordination with U^4+^.

Clustering analysis. With the same cutoff value, the conformational clustering analysis gave a significantly smaller number of clusters for the U^IV^-*t*-HOPO complex than that for *t*-HOPO^4−^, indicating that *t*-HOPO^4−^ was confined in a limited conformational space upon binding with U^4+^. To analyze the details in the conformational changes in the U^IV^-*t*-HOPO complex, a smaller cutoff value of 0.05 nm was used in the analysis of the U^IV^-*t*-HOPO complex to obtain a distribution of the clusters with the population of the cluster 1 (51%, Figure 12c) comparable with that of unbound *t*-HOPO^4−^ ligand. Consistently, the atomic positional RMSD of the U^IV^-*t*-HOPO complex (Figure 12), which fluctuated in the range of 0.02–0.12 nm, was much less than that of *t*-HOPO^4−^ (from 0.1 nm to 0.4 nm).

In addition, as seen from the time evolution of RMSD (Figure 12a), there was a conformational transition of the U^IV^-*t*-HOPO complex at about 50 ns, as seen from the sharp change in RMSD (Figure 12a), and during the last 50 ns, the simulation sampled the most populated cluster (cluster 1 with a population of 51.1%, Figure 12c and Appendix A). The superposition of the top three clusters (Figure 12b) showed noticeable changes in the positions of ligating water molecules and the *m*-HOPO rings neighboring to the water ligands, implicating their mutual influence.

Binding free energy. The binding free energy of U^IV^(κ^8^-*t*-HOPO) was calculated according to the following thermodynamic cycle:

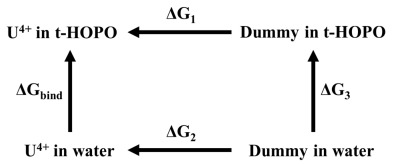


where ΔG_3_ represents the energy cost to transfer a dummy atom from water to HOPO and is equal to zero. Following the thermodynamic cycle, the binding free energy may be calculated as
 ΔG_bind_ = ΔG_1_ − ΔG_2_
where ΔG_1_ and ΔG_2_ were calculated by using the free energy perturbation (FEP) method.

As seen in Table 2, *t*-HOPO^4−^ showed strong affinity for U^4+^, which was calculated to be up to −81.3 kcal/mol. The strong interaction between *t*-HOPO^4−^ and U^4+^ benefitted from the strong electrostatic interaction between them. In addition, as discussed above, the *t*-HOPO^4−^ showed much lower conformational flexibility than its neutral counterpart *t*-HOPO^0^ as a consequence of the balance between the hydrophilic and hydrophobic interactions. The negatively charged *t*-HOPO^4−^ adapted better in the aqueous phase than *t*-HOPO^0^ and could be populated mainly with its dominant conformation (cluster 1, Figure 2 bottom). The Na^+^ in the vicinity of its *m*-HOPO rings helped to lock them in a conformation parallel to each other. It is interesting to note that in the dominant conformation of *t*-HOPO^4−^, the spermine backbone folded, and the four *m*-HOPO rings were on the same side (Figure 2), which was a conformation prepared to clamp a cation together. This was different from the main conformation of *t*-HOPO^0^, in which the two pairs of *m*-HOPO rings were on the opposite sides of the spermine backbone. The well-prepared conformation of *t*-HOPO^4−^ saved energy when clamping U^4+^ and could easily rearrange itself to adopt a conformation fitting with the U^4+^ ion.

The decomposition of the interactions between U^IV^ and *t*-HOPO^4−^ and between U^IV^ and water ligands (Table 2) showed much stronger electrostatic interactions between U^4+^ and *t*-HOPO, which was more than five times stronger than that between U^4+^ and water ligands. This stabilized the compact structure of the U^IV^(κ^8^-*t*-HOPO) complex.

## 3. Computational Detail Methodology

All molecular dynamics simulations were conducted by employing the Gromacs 5.1.4 package [30]. The bonded and non-bonded parameters of neutral and deprotonated HOPO were generated by the Antechamber module [31] in the Amber16 package [32] using the GAFF force field [33]. RESP charge was employed to describe the coulombic interaction and calculated by the semi-empirical quantum chemistry (sqm) tool in the Antechamber module. Water molecules were treated using the SPC/E water model [34]. The 12-6 Lennard-Jones (LJ) parameters of U^4+^ were taken from Merz et al. [35]. Long-range electrostatic interactions were calculated by the smooth particle-mesh-Ewald method [36]. The cut-off of the electrostatics and LJ potential were set to 1.0 nm. All hydrogen bonds were constrained by the LINCS algorithm [37] to allow a time step of 1 fs during the simulations. Periodic boundary conditions were imposed to all three dimensions.

All simulation systems were first relaxed by 1000 steps of energy minimization (the steepest descent), equilibrated 300 ps for NVT and then for 500 ps at a constant pressure of 1 atm and temperature of 310 K (isothermal–isobaric conditions, NPT). The equilibrated model systems were then sampled at 310 K and 1 atm with the NPT ensemble for 100 ns for analysis. During the simulations, the temperature was kept constant using the v-rescale weak coupling scheme with τ_T_ = 2.0 ps [38] and the pressure using the Berendsen barostat with τ_P_ = 1.0 ps [39].

Atomic positional root-mean-square deviation (RMSD) analysis was performed with all atoms in *t*-HOPO, *t*-HOPO^4−^ and U^IV^-*t*-HOPO complex taken into account, and the central structures of the most populated clusters in the simulations were used as the reference to distinguish the transition between the most populated cluster and other clusters in each simulation. The time evolution of RMSD with the starting structure as the reference was also calculated and is provided in the Appendix A.

In the clustering analysis, the algorithm of Daura et al. was used [40]. Owing to the different flexibilities of the *t*-HOPO, *t*-HOPO^4−^ and U^IV^-*t*-HOPO complexes, using the same cutoff value in the analysis of the three model systems would result in the too-sparse distribution of conformations of *t*-HOPO (smaller cutoff) or too few clusters of U^IV^-*t*-HOPO (bigger cutoff), as shown in Appendix A. Therefore, to obtain the proper distribution of the conformations to assist our understanding of the sampled conformational space and the flexibility of three model systems, the cutoff was set to 0.35 nm for the *t*-HOPO, 0.18 nm for the *t*-HOPO^4−^ and 0.05 nm for the U^IV^-*t*-HOPO complex, respectively. The intra-molecular weak interaction analysis of *t*-HOPO and *t*-HOPO^4−^ was carried out by using the independent gradient model (IGM) [41] implemented in the Multiwfn 3.5 program [42].

The compactness of the complex was estimated by calculating its radius of gyration (Rg)
$$\mathrm{Rg}\;=\sqrt{\frac{\sum_{i}m_{i}r_{i}^{2}}{\sum_{i}m_{i}}}$$
where *m*_i_ is the mass of atom *i* located at distance *r*_i_ from the center of mass of the complex. The calculation included U^4+^ and *t*-HOPO^4−^ in the complex.

When calculating the angles between each pair of the four hydroxypyridinone rings (Planes 1, 2, 3 and 4, Figure 1) of *t*-HOPO in its neutral and deprotonated states, the two C atoms and one N atom on each hydroxypyridinone ring (denoted as *m*-HOPO, see Figure 1) were selected to define the plane (marked by a Δ sign in Figure 1).

The binding free energy of *t*-HOPO with U^4+^ was calculated by using the free energy perturbation (FEP) method [43]. U^4+^ was perturbed from the gas phase to the aqueous phase, and each perturbed state was controlled by two variables λ_vdW_ and λ_coul_ for the perturbation of van der Waals and Coulomb interactions between U^4+^ and the solvent/ligand, respectively. The increment in each perturbation factor was 0.05, and in total, there were 41 windows. Each window underwent an equilibration phase composed of two sequential 1 ns simulations under NVT and NPT ensembles, respectively, which was followed by a 100 ns production run for analysis. The Gibbs free energy change in the two states was calculated using the Bennett acceptance ratio method [28] based on the energy difference between these windows.

## 4. Conclusions

The conformational dynamics of *t*-HOPO in its neutral and deprotonated states and the dynamics of U^IV^-*t*-HOPO were studied using MD simulations.

In its neutral state, *t*-HOPO showed high flexibility, and there were frequent transition events among the conformational spaces accessible under simulation conditions. The four hydroxypyridinone rings of *t*-HOPO exhibited propensity to be parallel in pairs and benefitted from the weak π-π stacking and H-bond interactions between hydroxypyridinone rings, which were not sufficient to resist the perturbation from the solvent, resulting in frequent exchange between their parallel and anti-parallel conformations.

In contrast, in its deprotonated state (*t*-HOPO^4−^), its accessible conformational space was significantly smaller, and the propensity of the four hydroxypyridinone rings to stay in paralleled states in pairs was enhanced. The analysis of the trajectories indicated that, besides the π-π stacking and H-bond interactions between hydroxypyridinone rings, there were also Na^+^ ions to lock the pairs of hydroxypyridinone rings via electrostatic attractions between the Na^+^ and *t*-HOPO^4−^ ligand, which could resist the disturbance of the water solvent.

Upon binding with U^4+^, *t*-HOPO could immobilize the cation by its eight oxygen atoms coordinating with the cation. Meanwhile, there were two water molecules in the first shell of U^4+^, resulting in the U^4+^ ion being deca-coordinated. Conformational analysis showed that the whole complex displayed less flexibility than *t*-HOPO in its neutral and deprotonated states owing to the strong electrostatic attraction between U^4+^ and *t*-HOPO^4−^.

## Data Availability

Not applicable.

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
