# Peer review of "Folding Dynamics of 3,4,3-LI(1,2-HOPO) in Its Free and Bound State with U4+ Implicated by MD Simulations"

_molecules, 2022, doi:10.3390/molecules27238151_

Round 1

Reviewer 1 Report

This work aims at studying the conformational dynamics of t-HOPO in its neutral and deprotonated states (t-HOPO4-) and the dynamics of UIV-t-HOPO by MD simulations. The authors first simulated the conformational spaces of the t-HOPO and t-HOPO4- states solvated in aqueous phase, and found that t-HOPO has higher flexibility. Then upon binding with U4+, the whole complex displayed the shortened residence time of water molecules and strong interaction. The authors applied molecular dynamics simulations to study the dynamics of t-HOPO and its complexation with actinides in aqueous phase which improved our understanding on the interaction modes between t-HOPO and actinides. However, some issues are needed to be addressed or answered before this manuscript can accepted for publication.

1.     The object of this work is focused on the reaction behavior of U(IV) and octadentate hydroxypyridonate ligand 3,4,3-LI(1,2-HOPO), which shows high affinity toward actinide ions (including uranium) and great potential on actinide decorporation. However, the uranium in here is mean the uranyl ion. When taken up into the body, other forms of uranium, including U(IV), are eventually oxidized to hexavalent uranyl ions (UO22+, U(VI)). Thus, the authors need to provide more information to explain that the necessity of studying the reaction behavior of U(IV) and 3,4,3-LI(1,2-HOPO).

2.     In addition, the “introduction” is too simple to understand the necessity of this work. The authors should complement the chemical form of uranium in vivo, as well as other applications of U(IV).

3.     A brief comparison of complexing capability and selectivity of 3,4,3-LI(1,2-HOPO) toward actinide ions will make more convictive than before.

4.     Is a wrong structure of 3,4,3-(LI-1,2-HOPO) provided in this work?

5.     In view of the time evolution of the angles between the four m-HOPO rings in Figure 3b, the angle between Plane1 and Plane2 was about 60°, contradicting the text "the four planes stayed parallel in pairs during the whole simulations". Please explain it.

6.     Please mark a or b in Figure 5 and Figure 9.

7.     Please go through the manuscript again and check carefully for expression, spelling, and grammar mistakes as well as typos.

Author Response

This work aims at studying the conformational dynamics of t-HOPO in its neutral and deprotonated states (t-HOPO4-) and the dynamics of UIV-t-HOPO by MD simulations. The authors first simulated the conformational spaces of the t-HOPO and t-HOPO4- states solvated in aqueous phase, and found that t-HOPO has higher flexibility. Then upon binding with U4+, the whole complex displayed the shortened residence time of water molecules and strong interaction. The authors applied molecular dynamics simulations to study the dynamics of t-HOPO and its complexation with actinides in aqueous phase which improved our understanding on the interaction modes between t-HOPO and actinides. However, some issues are needed to be addressed or answered before this manuscript can accepted for publication.

  1. The object of this work is focused on the reaction behavior of U(IV) and octadentate hydroxypyridonate ligand 3,4,3-LI(1,2-HOPO), which shows high affinity toward actinide ions (including uranium) and great potential on actinide decorporation. However, the uranium in here is mean the uranyl ion. When taken up into the body, other forms of uranium, including U(IV), are eventually oxidized to hexavalent uranyl ions (UO22+, U(VI)). Thus, the authors need to provide more information to explain that the necessity of studying the reaction behavior of U(IV) and 3,4,3-LI(1,2-HOPO).

Response1: We acknowledge the suggestion of the reviewer. We agree with the reviewer that uranyl is the most stable form of uranium in oxidative environment. The reason to study the behavior of 3,4,3-LI(1,2-HOPO) with U(IV) is as below:

(1) the present work represents part of our study on the interaction of 3,4,3-LI(1,2-HOPO) with actinides. We are considering key actinides in their representative oxidation states, specifically for uranium, in both its tetravalent and hexavalent states. The study of uranyl is more complex due to the structure feature of the ligand, which is octa-dentate distributed on four rings that are bonded sequentially on a spermine chain. This structure feature causes the coordination of the ligand with uranyl, which is a linear tri-atomic ion, flexible, and the simulation study of this system is still undergoing. We will share the results with the community once available.

(2) the tetravalent uranium is stable with 3,4,3-LI(1,2-HOPO) coordinated. In previous studies, tetravalent uranium has been identified as one of the main species of in vivo uranium. Though we can say that it would eventually oxidized to uranyl in oxidative environment, we don’t know its lifetime after being taken into the body. In addition, in a recent work, Abergel group reported the observation of stable UIV(3,4,3-LI(1,2-HOPO)) complex at physiological pH values. This shows the necessity to understand the behavior of this complex in aqueous phase to obtain insight on the decorporation of in vivo U(IV) by 3,4,3-LI(1,2-HOPO).

A sentence was added to the end of the “Introduction” section to mention this issue:

Its complexation with tetravalent uranium (U4+) was also investigated here to offer insights on the influence of its complexation with monatomic cations in aqueous phase. In early studies, tetravalent uranium has been shown to be stable under physiologic conditions, and its coordination with t-HOPO enhances its stability in aqueous phase,[27] while the conformation and dynamics of the complex remain lack of molecular level of understanding.

  1. In addition, the “introduction” is too simple to understand the necessity of this work. The authors should complement the chemical form of uranium in vivo, as well as other applications of U(IV).

Response2: This work intends to report our study on the folding dynamics of t-HOPO in its native state and in its deprotonated states, and its complexation with U(IV) was chosen as a case study to show the influence of coordination with cation on its folding dynamics.

Regarding the in vivo speciation of uranium, survey of literature shows that earlier studies didn’t put much attention on the chemical form of in vivo uranium, and in general terms uranium was considered to exist in its tetra- and hexa-valent states with water, carbonate, and/or protein (e.g. transferrin, siderocalin, etc.) coordinated. This indicates that the in vivo speciation and dynamics of uranium remain to be studied. At the aware of this situation, we are applying molecular dynamics simulations to study the in vivo speciation and dynamics of uranium to obtain molecular level of understanding. We will share our finding in the future when ready.

Following the suggestions of the reviewer, we modified the last paragraph of the Introduction section to mention the motivation to study the UIV(t-HOPO) complex.

  1. A brief comparison of complexing capability and selectivity of 3,4,3-LI(1,2-HOPO) toward actinide ions will make more convictive than before.

Response3: We acknowledge the suggestion of the reviewer. This work mainly focuses on the folding equilibria of 3,4,3-LI(1,2-HOPO) in the native and its deprotonated states. The results with U(IV) bound are reported here to show its influence on the behavior of 3,4,3-LI(1,2-HOPO) in aqueous phase. We will report our work on the interaction between 3,4,3-LI(1,2-HOPO) and other key actinide ions in a separate manuscript which is under preparation.

  1. Is a wrong structure of 3,4,3-(LI-1,2-HOPO) provided in this work?

Response4: We acknowledge the careful reading of the reviewer, and have corrected the elucidation of 3,4,3-LI(1,2-HOPO) in the manuscript.

  1. In view of the time evolution of the angles between the four m-HOPO rings in Figure 3b, the angle between Plane1 and Plane2 was about 60°, contradicting the text "the four planes stayed parallel in pairs during the whole simulations". Please explain it.

Response5: The figure shows that the angle mainly populates in the range of 0 – 40 degrees with the peak at around 10 degree. To avoid misunderstanding, the sentence is modified as below:

Meanwhile, in view of the time evolution of the angles between the four m-HOPO rings in Figure 3b, the four planes show high propensity to stay parallel in pairs during the simulations.

  1. Please mark a or b in Figure 5 and Figure 9.

Response6: The labels were added in Figure 5 and Figure 9 (Figure 10 in new manuscript) as suggested.

  1. Please go through the manuscript again and check carefully for expression, spelling, and grammar mistakes as well as typos.

Response7: We have read through the manuscript and corrected the problems we found.

Reviewer 2 Report

This meaningful manuscript contains many useful information about the dynamics of t-HOPO and its interactions with U4+. Thorough computational analysis was conducted, and insightful finding were reported. Therefore, I would recommend publish it with minor revision.

1.    The introduction part seems to be too concise (only 21 lines) and many relevant literatures are missing. For example, the author should mention the prior literatures about complexation of other promising decorporation agent with actinides. Besides, the author should also mention the research status of the dynamics of the other promising decorporation agents.

2.    In the computation detail and methodology part, the decomposition of the simulation boxes should be mentioned. The author should inform readers how many U4+, ligand and other species in the simulation box.

3.    Most importantly, the author should validate the simulation before analyzing simulation results. For example, can the density of your simulation box agree with experiments? Can the simulation predict the correct dynamic properties (viscosity, diffusion coefficient, et al)?   

4.    In clustering analysis, the author used a cutoff of 0.35 nm for t-HOPO. The author should mention why this number is a good choice.

5.    In line 197-199, the author mentioned “These wandering Na+ ions did not move far away, but migrate back quickly” The author should provide evidence for this argument. What’s the RMSD of Na+ ? 

Author Response

This meaningful manuscript contains many useful information about the dynamics of t-HOPO and its interactions with U4+. Thorough computational analysis was conducted, and insightful finding were reported. Therefore, I would recommend publish it with minor revision.

  1. The introduction part seems to be too concise (only 21 lines) and many relevant literatures are missing. For example, the author should mention the prior literatures about complexation of other promising decorporation agent with actinides. Besides, the author should also mention the research status of the dynamics of the other promising decorporation agents.

Response1: We acknowledge the suggestion of the reviewer, and the introduction has been modified.

  1. In the computation detail and methodology part, the decomposition of the simulation boxes should be mentioned. The author should inform readers how many U4+, ligand and other species in the simulation box.

Response2: The composition of the simulation box for UIV-t-HOPO complex has been added in Table SI of the Supporting Information.

  1. Most importantly, the author should validate the simulation before analyzing simulation results. For example, can the density of your simulation box agree with experiments? Can the simulation predict the correct dynamic properties (viscosity, diffusion coefficient, et al)?   

Response3: In this work, the water molecules were treated by SPC/E model, and the parameters of U4+ were taken from Merz et al, adapted to the SPC/E model (J. Phys. Chem. B 2015, 119, 883-895). The parameters of HOPO were adopted from the library of AMBER force field. This strategy offers us with confidence on the treatment of pair interactions in the model systems. For the molecular dynamic simulations, this work follows the well tested simulation protocols, i.e. energy minimization, then equilibration, then production run. During the equilibration, the systems were equilibrated with the NVT ensemble followed by NPT ensemble, after which another NPT run was conducted for analysis. This guarantees the simulation under 1 atm with correct density (1.0 g/cm3, which is close to neat water) and reasonable diffusion coefficient of water at simulated temperature. The data used for analysis were obtained from the final production run with the NPT ensemble.

  1. In clustering analysis, the author used a cutoff of 0.35 nm for t-HOPO. The author should mention why this number is a good choice.

Response4: The t-HOPO is very flexible compared to its deprotonated state, which can be expected. During the analysis, we have tried a few values, and the value of 0.35 nm could offer us appropriate evaluation of the conformational space of the native ligand sampled during the simulations.

  1. In line 197-199, the author mentioned “These wandering Na+ ions did not move far away, but migrate back quickly” The author should provide evidence for this argument. What’s the RMSD of Na+?

Response5: As RMSD is not a good measure to evaluate the migration of Na+ ions, we calculated their distance with the nearest atom of HOPO rings, and the figure is shown below. We can see that during the simulations, the four Na+ ions move back and forth to interact with negatively charged t-HOPO4-. This figure was also added to the manuscript (Figure 7).

Round 2

Reviewer 1 Report

The authors provid detailed response and modification based on the comments, I suggest publication in its current version.

Author Response

We acknowledge the positive comment of the reviewer.